# Farms or Forests? Understanding and Mapping Shifting Cultivation Using the Case Study of West Garo Hills, India

**Amit John Kurien [1,2,*]**, **Sharachchandra Lele [1]** and **Harini Nagendra [3]**

1   Centre for Environment and Development, Ashoka Trust for Research in Ecology and the Environment (ATREE), Royal Enclave, Jakkur P. O., Bengaluru, Karnataka 560064, India
2   Manipal Academy of Higher Education (MAHE), Manipal, Karnataka 576104, India
3   School of Development, Azim Premji University, PES Institute of Technology Campus, Pixel Park, B Block, Electronics City, Hosur Road, Bengaluru, Karnataka 560100, India
*   Correspondence: amitkurien@gmail.com; Tel.: +91-80-23635555

**Abstract:** Attempts to study shifting cultivation landscapes are fundamentally impeded by the difficulty in mapping and distinguishing shifting cultivation, settled farms and forests. There are foundational challenges in defining shifting cultivation and its constituent land-covers and land-uses, conceptualizing a suitable mapping framework, and identifying consequent methodological specifications. Our objective is to present a rigorous methodological framework and mapping protocol, couple it with extensive fieldwork and use them to undertake a two-season Landsat image analysis to map the forest-agriculture frontier of West Garo Hills district, Meghalaya, in Northeast India. We achieve an overall accuracy of ~80% and find that shifting cultivation is the most extensive land-use, followed by tree plantations and old-growth forest confined to only a few locations. We have also found that commercial plantation extent is positively correlated with shortened fallow periods and high land-use intensities. Our findings are in sharp contrast to various official reports and studies, including from the Forest Survey of India, the Wastelands Atlas of India and state government statistics that show the landscape as primarily forested with only small fractions under shifting cultivation, a consequence of the lack of clear definitions and poor understanding of what constitutes shifting cultivation and forest. Our results call for an attentive revision of India's official land-use mapping protocols, and have wider significance for remote sensing-based mapping in other shifting cultivation landscapes.

**Keywords:** jhum; swidden; shifting cultivation; land use mapping; wasteland; social construction; forest classification; Forest Survey of India; Meghalaya; India

## 1. Introduction

Shifting cultivation (also known as swidden) is a widely practiced form of agriculture important for livelihood, nutrition and as a safety net for millions of people in the tropics [1,2]. It also arguably contributes to biodiversity conservation, soil and water conservation, and climate change mitigation [3–5]. Simultaneously, however, shifting cultivation has been criticized by researchers and policy makers alike for being environmentally destructive, and is often referred to as wasteland in government documents [6–8]. Across the world, the understanding of the extent and form of shifting cultivation, how much it is intensifying, and whether it actually causes deforestation, remains limited [8]. A large proportion of the world's forest-agricultural frontier is still occupied by shifting cultivation [9]. The persistence of this practice in the face of focused efforts to eradicate and replace it, as well as pressures of population increase and market penetration has elicited scholarly interest.

Moreover, shifting cultivation could be a vital ingredient for ensuring forest cover, nutritive health and livelihood development in regions where infrastructure development for modern agriculture is a fractious choice [10,11].

South Asia is underrepresented in the global shifting cultivation literature (see [2]), although it is widespread in Northeast India and exists in several other locations [12–14]. Northeast India, where such cultivation is officially known as jhum, has also seen it being both celebrated for its contribution to safeguarding livelihoods [12,15] and reviled as the cause of deforestation [16,17]. Provincial governments have therefore consistently curtailed jhum and sought to encourage cash crop plantations [18–20]. The pioneering work on jhum in the region focused on understanding its forms and field-level agro-ecology [15]. Estimates of the extent of shifting cultivation in the landscape and its trends, however, began only in recent times with the emergence of remote sensing. Unfortunately, the literature shows wide discrepancies and fluctuations in these estimates. For instance, Talukdar, et al. [21] claims the extent of shifting cultivation in Garo hills to be just 500 km$^2$ out of a total landscape of 8167 km$^2$, while Behera, et al. [22] identified no class as shifting cultivation, but mapped fallows and identified 281 sq. km area as wasteland in 2005 in 280,288 km$^2$ of Northeast India. These differences, as we shall substantiate, are rooted in varying and unsuitable definitions of shifting cultivation and forest land-uses and their corresponding land-covers, and inadequacies in image interpretation and verification. Moreover, following improved definitions and interpretation, if one can extract some information on aspects such as the duration of fallow periods and its correlates, such remote sensing-based studies would make an additional contribution to understanding the drivers of changes in shifting cultivation.

We present a case study that seeks to make both methodological and empirical contributions to the mapping of shifting cultivation landscapes. Methodologically, we draw upon the distinction between land-use and land-cover [23] and outline the complex land-cover changes that shifting cultivation as a land-use passes through. It is therefore necessary to rigorously define what constitutes active shifting cultivation and fallow and what constitutes forest, which in turn requires detailed field data collection [24]. Empirically, the application of this approach provides the first reasonable estimates of the actual extent of shifting cultivation and forests and of the average duration of the shifting cultivation cycle in the Garo Hills region. Through this work, we highlight how the choice of class definitions and methods make certain land-use classes invisible and others magnified on a landscape and its implications for science and policy.

## 2. Defining, Mapping and Analyzing Land-Use/Land-Cover in Shifting Cultivation Landscapes

A review of literature on the mapping of shifting cultivation and forests, as provided below, helps us to understand what the purposes of mapping shifting cultivation are, and therefore what classification schemes are needed, as well as what is currently practiced in the literature. This sets the background and provides rationale for our study.

Shifting cultivation is a hill-based agricultural system involving the clearing and burning of natural vegetation, followed by the cultivation of new fields for a few years. This is followed by a period of fallow during which the vegetation regenerates, after which the cycle begins all over again. There are many types of shifting cultivation [25], but in most of them, the fallow periods were traditionally a few decades long, enabling significant biomass regrowth that reached a secondary forest form before being cleared and cultivated again. In this study, we define shifting cultivation as consisting of both the actively cultivated phase, as well as the fallow phase. Much of the literature on shifting cultivation has been motivated by two distinct societal concerns. Conservationists have been concerned that shifting cultivation causes deforestation [16,26,27]. On the other hand, development policy makers have considered shifting cultivation to be a primitive, unproductive form of agriculture [7,28] that therefore requires development [29]. Although these extreme positions have been moderated somewhat [30,31], the debate continues. One of the central concerns now is that shifting cultivation

may be intensifying, resulting in shorter fallow periods, reducing the biomass regrowth and changing its species composition, making cultivation unproductive and unsustainable [2,32,33].

In order to establish whether and where either of these processes—deforestation (net reduction in natural forest) and/or intensification (shortening of the fallow period)—are happening, it is necessary, firstly, to clearly define what comprises shifting cultivation and forest land-uses. Secondly, it is necessary to identify what other land-use classes in the landscape might be confused with shifting cultivation or forest in terms of land-cover, from this generating the classification protocols needed to distinguish them. Thirdly, the spatial patterns of active shifting cultivation and fallow fields could help in understanding the nature of the shifting cultivation cycle prevailing. We summarize the literature on each of these steps, beginning with the global literature and then providing an overview of the literature on south Asia.

## 2.1. Classifying Shifting Cultivation Landscapes

The literature on land-cover/land-use classification generally subscribes to the notion of objective and universal classification schemes (see [34,35]). But the literature in critical geography has pointed out that classification schemes are necessarily social constructs, reflecting distinctions that are valued by the classifying agent [36–40]. A land-use map would therefore be more useful if it reflected categories of importance to multiple stakeholders [36,38,41]. The first step in land-use classification should therefore be to determine which classes are most pertinent to which social or policy concerns in a given context [42].

For instance, if forests are considered important because they are repositories of high biodiversity, then the forest class must generally contain higher biodiversity than the non-forest class (es). In such a case, it would be inappropriate to include single-species tree plantations in the forest class or to merge shifting cultivation fallows into forest. If, however, forests are valued only for their sequestered carbon, then it would be appropriate to include all high-carbon forms (including old-growth forest, old fallows, and single-species tree plantations) under forest. If the same land-use map is to be useful to different audiences with different interests—conservation, agricultural productiveness, carbon sequestration potential and/or sustainability questions in shifting cultivation—the map must make adequate distinctions between all of these classes: high/low biodiversity land-uses, old/young fallows and high/low carbon land-uses.

Much of the global literature on shifting cultivation—dominated by studies from Southeast Asia, with a few from Madagascar, Mozambique and South America—has internalized this concept to some extent. Most studies distinguish between active shifting cultivation fields and settled cultivation (e.g., wet rice cultivation) [43–45]. Many also distinguish between old shifting cultivation fallows and secondary/relatively undisturbed forest [46,47]. Some distinguish between old-growth forests and single-species tree plantations [48,49], while others (typically those using NDVI approaches for classification) do not clearly do so [50].

## 2.2. Mapping Shifting Cultivation Stages, Forests, and Forest-Like Land-Uses

The next challenge is translating land-cover detectable in satellite imagery into the socially-relevant land-use classes identified. While this is a generic issue in land-use mapping, it is particularly challenging when mapping shifting cultivation landscapes because shifting cultivation cycles through a whole range of land-cover classes: cleared land, burned land, land covered with crops, post-harvest land, fallows that start as abandoned fields with scrub/grass, become bush-covered and end up as secondary forest. Consequently, a land-cover class such as shrubs, scrub or bush may be interpreted as degraded forest in a context where shifting cultivation does not exist, but likely represents young fallows in shifting cultivation landscapes. Similarly, grassy patches on slopes may be classified as forest blanks by foresters, but in a shifting cultivation landscape may represent the second or third year cropping fields.

Some researchers have addressed the challenge of continuously changing land-cover by mapping shifting cultivation landscapes, i.e., those areas showing continuously changing land-cover (where patches cycle between cleared and fallow phases), separately from the other land-use classes of stable-high and stable-low biomass land-covers [43]. While this enables mapping of the total area under the shifting cultivation cycle, it does not help us estimate the fallow versus active cultivation areas. Others [50–52] have used the burned fields or cleared fields [53] to distinguish active shifting cultivation plots, but in areas where active cultivation continues into the second or third year without burning (only clearing), this technique leads to an underestimation of the area under active cultivation and an over-estimation of fallows. The landscape mosaic map approach proposed by Messerli, Heinimann and Epprecht [51] altogether bypasses the step of inferring land-use from land-cover for each pixel or plot by constructing land-cover mosaics, a contiguous set of pixels that have similar associations of land-cover in a large window around them. They then further generate a typology of landscape mosaics from these land-cover mosaics that seem to represent areas of land-use intensification and/or forest degradation that may be relevant for policy makers. This approach, while valuable from the perspective of identifying different land-use dynamics in a large (province or country-scale) landscape, does not help to identify the extent of active and fallow shifting cultivation in a specific region.

To clarify the challenge further, consider the schematic diagram in Figure 1. that shows three plots at three different stages in the shifting cultivation cycle (typical of the study region). The 0th year plot, which is the oldest fallow, enters active cultivation at the end of the agricultural year when it is cleared and burned. The 1st year plot is cultivated and harvested, then cleared and prepared for a 2nd year of cultivation. The 2nd year plot is also cultivated and harvested but then abandoned or fallowed for the long run. The area under active cultivation in a given year would be the area in either of the two dashed boxes in the Figure 1 but not the area of all three plots. Methods that use single images from the post-burn phase and map only burned plots run the risk of missing half the active area in a region where the 1st year plot is only cleared and not burned for the 2nd year of cultivation. On the other hand, if one tries to include both burned and cleared fields from the post-clearance & post-burn phase image, one risks including the now-fallowed 2nd year field also, thereby over-estimating the active cultivation area.

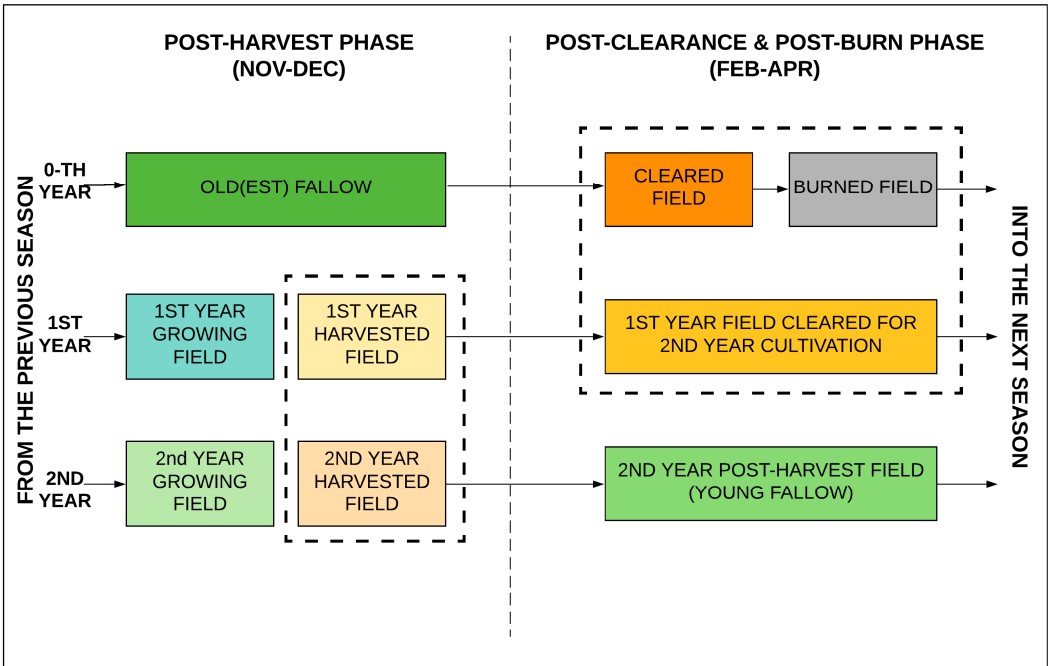

**Figure 1.** Schematic representation of the sequential changes in land-covers under active shifting cultivation (ASC). Dashed boxes represent two alternate ways of measuring total area under active cultivation.

From this diagram, it appears that using imagery from the post-harvest (November–December) period might avoid both under and over-estimation. There are, however, several reasons why a single post-harvest image may not be sufficient. First, the period immediately after harvest (October–December) tends to have high cloud cover, particularly in the South Asian context. So although shifting cultivation harvested fields may be visible, other separating other classes maybe tough. Second, if the purpose of land-use mapping is also to distinguish between other classes (such as plantations and secondary or old-growth forest) then imagery from the dry season (February–April) is often superior to imagery from other seasons, as phenological differences in tree cover are more visible then (Figure S1 in Supplementary Materials). In short, as Hurni, et al. [54] rightly argue, multi-season imagery is preferable for mapping shifting cultivation landscapes as a whole (although cloud cover may be a constraint). Along with the dynamic nature of land-cover within the shifting cultivation cycle is the fact that many stages in the cycle resemble land-covers under other land-uses [8,55]. Active shifting cultivation may resemble settled cultivation, and older fallows may resemble old-growth forests. Moreover, the introduction of perennial tree crops such as cashew, rubber or areca complicates matters further. This strengthens the argument for multi-season data and extensive ground-truth to enable proper land-use classification.

### 2.3. Mapping to Identify Spatial Patterns in Fallow Periods/Land-Use Intensity of Shifting Cultivation

An expansion in the total area under the shifting cultivation cycle would lead to deforestation (defined as a net reduction in old-growth forest area). But if additional area is unavailable, shifting cultivation may intensify by a shortening of the fallow period, leading possibly to the unsustainability of production. Hence, estimation of fallow periods or land-use intensity of shifting cultivation, trends in them and the drivers of these trends is of interest to policy-makers.

A global-level meta-analysis of case studies has already indicated shortening of fallow periods in many parts of the world [2]. But such estimates of fallow period and identification of its trends come from detailed field studies encompassing a few villages [56,57]. Remote-sensing based analyses of fallow periods are few. Hett, et al. [53] use a moving window technique to identify five landscape mosaics of different crop-fallow cycle intensities and their spatial and temporal distribution in Lao PDR. More recently, Dutrieux, et al. [58] used a Landsat time-series of Normalised Difference Moisture Index (NDMI) datasets to delineate fallow periods, and Jakovac, et al. [59] have used remote sensing to estimate fallow periods and patterns in intensification in Amazonia. Messerli, Heinimann and Epprecht [51] also identify intensification in the landscape, but their definition of intensification is a transition from shifting to settled agriculture, not specifically a shortening of the fallow periods. The drivers of declining fallow periods have been debated extensively, ranging from population alone [60] to multiple drivers and complex pathways [61]. Analyses of drivers of agricultural change and deforestation using long time-series satellite exist for several parts of the world such as Africa and Amazonia.

### 2.4. Shifting Cultivation Mapping in South Asia

While the international literature is to an extent engaged with these challenges of definition, interpretation, and estimating intensification, the South Asian literature has by and large lagged behind on all fronts. First, the official Indian forest mapping agency, the Forest Survey of India (FSI), has consistently defined forest cover as all tree cover above 10% tree canopy density [62], thereby including horticultural plantations like rubber or tree-shaded crops like coffee in its estimates of forest cover. Some researchers in this region have also followed this approach [63,64], but a cursory visit to the region suggests that it will result in over-estimation of natural forest cover due to the ubiquity of regrowing fallows and horticultural plantations. Others that focus on mapping floristic classes of forest do, however, distinguish between natural forest and plantations [22]. But such forest-focused studies ignore shifting cultivation as a land-use, thereby implicitly including different phases of shifting cultivation in the different forest or non-forest classes. For instance, Behera, et al. [22] distinguish

16 classes in their land-use change study covering 1985 to 2005. They define a fallow class and a wasteland class without clarifying what it contains, and do not have a separate class for active shifting cultivation or its cyclical fallows. Not surprisingly then, claims about shifting cultivation as the cause of deforestation made from studies that only map forest cover and not shifting cultivation [65] do not hold water.

Second, where shifting cultivation is explicitly part of the classification, its treatment is confusing or inadequate. For instance, Roy et al. [66] use current jhum, abandoned jhum and shifting cultivation to represent the practice. The Wastelands Atlas of India [67], the only official source that maps shifting cultivation, maps only the burned and cleared areas, excluding the multiple cultivation years, but terms it a category of wasteland, indicating the bias against shifting cultivation as a legitimate land-use.

Third, the confusion between land-cover and land-use and the poor choice of classes is often compounded by limited ground-truth and post-classification accuracy assessment. For instance, in mapping land-cover for the whole state of Meghalaya in Northeast India—an area of 22,429 km$^2$—Roy and Tomar [68] use only 69 pixels for accuracy assessment for a total of 10 classes, while others [21,69] provide no details on ground-truth used. Singh, et al. [70] and Roy and Joshi [71] acknowledge the need to map jhum and abandoned jhum, but it is not clear how 277 ground-truth points in all (in the latter study) suffice for 13 classes in a region of 255,134 km$^2$ and with 188 m spatial resolution. Overall, there is a bias towards tree-focused classification and a lack of shifting cultivation-relevant methods in remote-sensing based studies. Finally, studies that examine trends in fallow periods are limited only to village-level investigations [72]. Landscape-level analysis of the correlates of changing fallow periods has not yet been attempted in this region.

## 3. Approach and Objectives

In light of the review of the literature provided above, our study adopted the following approach and objectives:

1.  Given that the debate on shifting cultivation is driven by both biodiversity concerns and agricultural productivity/sustainability concerns, we seek to distinguish between (secondary forest-like) old fallows, old-growth forest, and horticultural tree plantations, and also between active shifting cultivation fields and wet rice valley cultivation, and between young and old fallows.
2.  Given the similarities with other (non-forest and non-shifting cultivation) land-uses, we seek to demonstrate the importance of using two-season data and substantial ground truth to achieve such a separation.
3.  Given the concern about possible declines in fallow periods, we propose the use of fallow: active shifting cultivation ratios to estimate the fallow duration in different sub-regions and the possibility of relating the variation in these ratios across the landscape with other land-uses in and demography of these sub-regions.
4.  Applying these methodological improvements to a site in Northeast India, we estimate the extent of active and fallow shifting cultivation, old-growth forest and other land-uses in that region, and compare our findings with existing estimates to highlight our empirical contribution. We also identify possible correlates of shifting cultivation intensities and their implications.

## 4. Study Area and Methods

### 4.1. Study Area and Major Land-Uses

The study area is the district of West Garo Hills in the state of Meghalaya in Northeast India. It lies between 25°47′ to 26°10′ N latitude and 89°45′ to 92°47′ E longitude (Figure 2). The region is characterized by an undulating terrain (ranging from 15 m at the border with Bangladesh up to 1400 m ASL). The climate is humid sub-tropical at lower elevations and sub-temperate in the upper hills, with average annual rainfall ranging from 2000–4500 mm. The district has an area of 3677 km$^2$ with an overall population density of 175 persons per km$^2$ [73] and is divided into eight sub-districts or

community and rural development (CRD) blocks (Figure 2). The district was subdivided into West and South-West Garo Hills districts in 2012; however, we have mapped the erstwhile West Garo Hills for the sake of comparability with other data.

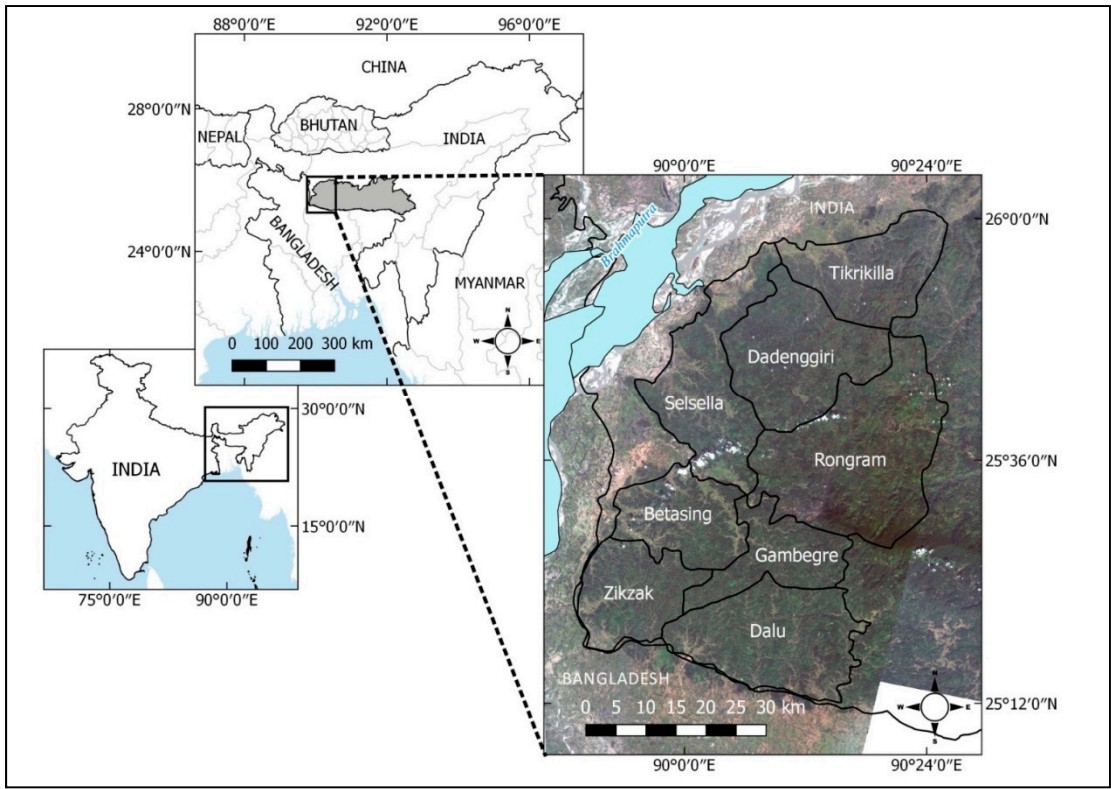

**Figure 2.** The erstwhile West Garo hills district in the state of Meghalaya (grey area in middle inset) in Northeast India.

The natural vegetation of this region is semi-evergreen and wet evergreen forest. As in other parts of the state and the region, however, shifting cultivation is ubiquitous in the landscape. The main crops include the many varieties of hill rice (*Oryza sativa*), maize (*Zea mays*), millets, along with a large variety of vegetables, tubers and leafy greens. The land is typically cultivated for two years before fallowing. The first-year fields are created by clearing (Figure 3a) and burning (Figure 3b) an old fallow in the month of March or April, and sown immediately after (Figure 3c) in anticipation of the rains. After the first year harvest in November, the field is cleared (Figure 3d) to prepare it for another year of cultivation. Once abandoned (usually after the 2nd year of cultivation), the fields may remain fallow for anywhere between 1 to 20 years, although most plots we visited were re-cultivated within 4 to 5 years, and fallows older than 12 years were rare (see Figure 3c for a view of a 7-year fallow).

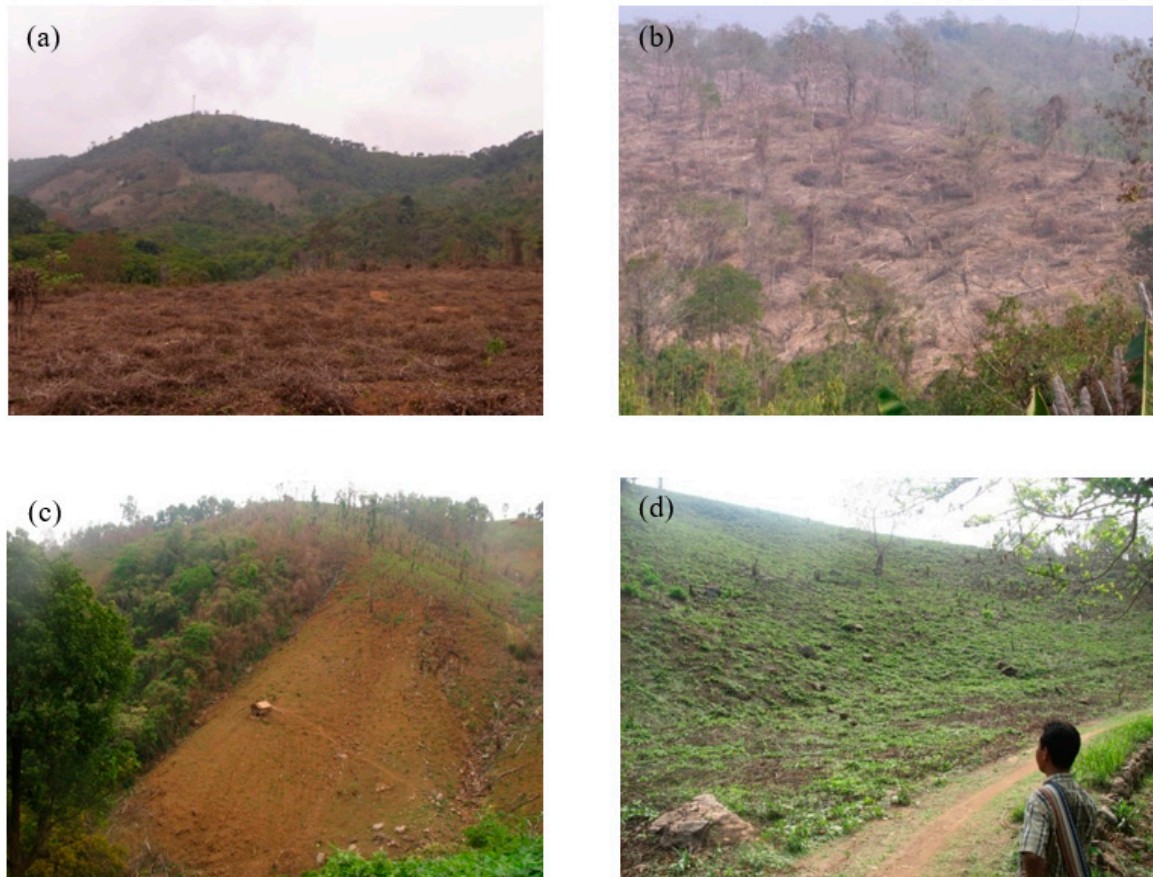

**Figure 3.** Shifting cultivation land-covers in West Garo Hills: (**a**) cleared field before burning, (**b**) burned field in late February, (**c**) first year field post clearing, burning, and sowing in May (with 7-year-old fallow to the left), (**d**) second year active cultivation field after clearing in May.

The other main land-uses in the hills include old-growth forest (i.e., relatively undisturbed or mature secondary regrowth forest), and private monoculture plantations of cashew (*Anacardium occidentale*), areca or betel palm (*Areca catechu*) and rubber (*Hevea brasiliensis*), along with other smallholder mixed plantations of oranges, tea, coffee and banana. Low-lying areas in the valleys usually have wet rice agriculture, with some home gardens and small-scale seasonal vegetable farming.

In this context, we chose the main land-uses to be distinguished as those relevant to concerns about shifting cultivation extent and intensity, about other agricultural land-uses, and about natural forest cover:

1. Active shifting cultivation (first and second year)
2. Young fallow (1–10 years fallow period or 3–12 years post-burning)
3. Old fallow (11–20 years fallow period or 13–22 years post-burning)
4. Old-growth forest (>20 years fallow period)
5. Horticultural plantations (cashew, areca, rubber)
6. Wet rice (valley) cultivation
7. Other (mixed/home garden) cultivation
8. Water bodies

The choice of years to distinguish between young and old fallows was based on earlier studies that suggest that a cultivation cycle of less than 10 years is likely to be unsustainable [15,74]. Similarly, after 20 years of fallow, the secondary forest is seen (locally) as being indistinguishable from old-growth forest (even the local name is the same; Table S1 in Supplementary Materials), and the biodiversity of

such fallows in the region approaches that of old-growth forest [75]; hence any forest not cultivated in the past 20 years was treated as old-growth.

*4.2. Methodology*

### 4.2.1. Choice of Imagery

In Meghalaya, the period between May and November is characterized by high cloud cover, and only a 4–6 month window starting mid-November is relatively cloud-free. This includes the post-harvest phase and the clearing and burning phases. For reasons already indicated in Section 2.2, we decided to use two-season data: one corresponding to the post-harvest phase and another to the clearing/burning phase. We purchased and tested Indian Remote Sensing Satellite LISS-IV imagery (5.8 m resolution). Eventually, however, we decided to use Landsat 8 Operational Land Imager (OLI) data (30 m resolution), because we found Landsat 8 to have more spectral information (eight bands, excluding thermal bands, as compared to three bands in LISS-IV) and a wider distribution of digital number (DN) values in each band. We used imagery for the agricultural year 2013–2014, viz., post-harvest data of 13 November 2013 and post-clearing and post-burn data of 22 April 2014. Path/row 138/042 covered more than 97% of the study area. The remaining area was filled in with images from the neighboring tile 137/042 for similar dates (22 November and 30 March) were used to complete the study area.

### 4.2.2. Ground Data Collection

The first author stayed in the study area between November 2013 and May 2014 and conducted extensive ground data collection across the post-harvest (winter) and post-clearance and post-burn (summer) seasons. Data were gathered from more than 16 villages spread across the district, as well as along main road networks across the study area. Village transects were traversed with a knowledgeable residents to understand the different stages of shifting cultivation. Ground data gathered for land-use polygons were always of a minimum of $90 \times 90$ m ($3 \times 3$ pixels of Landsat imagery) to minimize positional error [76,77]. GPS readings of each corner of the polygon were taken with an error ≤10 m. We augmented the ground data using high-resolution (sub-meter) Google Earth images to improve spatial coverage for certain classes. The spectral signature of each polygon was then compared with the average signature for its land-use class and outliers (beyond 2 standard deviations) were discarded. A total of 677 polygons including more than 45,000 pixels across 13 land-use classes were used in training and validation.

### 4.2.3. Image Processing, Classification and Validation

Image processing was conducted using ERDAS Imagine 9.1 software. One image was geo-rectified using a 1st-order polynomial and eight ground control points (GCPs), ensuring RMS error of less than one pixel. All other images were co-registered to this image, mosaicked, and clipped to the boundaries of West Garo Hills district obtained from Census of India maps. We omitted the thermal bands 10–12, band 9 that is meant for cirrus cloud detection, and the panchromatic band (band 8). The two-season 7-band images were then stacked into a single 14-band file. To reduce confusion created by tree vegetation surrounding houses, the main settlements were masked out. For classification, we used the maximum likelihood classification (MLC) algorithm with 367 polygons (averaging almost 30 polygons per land-use class) encompassing >25,000 pixels as the training data. Post-classification smoothing was carried out using a $3 \times 3$ pixel majority filter to remove speckling. Given the multiplicity of land-cover classes subsumed under the active shifting cultivation (ASC) land-use class (Figure 1), we had to use a split-and-aggregate strategy for accurately delineating this class. Active shifting cultivation was split into three classes corresponding to three different patterns of land-cover: fallow in November and cleared (not yet burned) in Feb-March (ASC-0Ycleared), fallow in November and burned in March-April (ASC-0Yburned), and harvested in November (year 1 cultivation) and then

cleared in April for year 2 of cultivation (ASC-1Y2Y) (See Figure 1). We consciously omitted patches that were harvested in November and abandoned by April to avoid double-counting as explained in Section 2.2. Accuracy assessment (validation) was done using 310 polygons that covered >19,000 pixels. However, to reduce spatial autocorrelation due to pixels originating in the same polygon, a 30% subset of randomly chosen pixels from each polygon was eventually used (amounting to 5808 pixels, with adequate pixels in each class). The distribution of training and validation polygons and pixels across classes is given in Table S2 (Supplementary Materials). The Kappa statistic [77] was used as an indicator of overall accuracy.

### 4.2.4. Estimating the Fallow Period

If one assumes that the shifting cultivation cycle is a stable one, then at any given point in time, the distribution of active shifting cultivation (ASC) and fallow plots on the landscape within which a particular cycle is being practiced would be similar to the time sequence through which any particular plot cycles. Therefore, the fallow period of that particular cycle can be estimated by:

$$\text{Fallow period} = n \times (\text{F:ASC ratio}) \tag{1}$$

where "*n*" is the typical number of years of consecutive cropping in an active field, and F:ASC is the ratio between the area under fallow (young and old) and the area of ASC in that cycle. This method could be used at multiple scales: individual cycles, entire villages containing multiple cycles, or regions containing multiple villages. The main assumption is that one can somehow identify the spatial boundary of the cycle and that all plots in a particular boundary are part of the same cycle. In the study region, villages carry out shifting cultivation within certain traditional boundaries, and in general, there is little spatial overlap between cycles of different villages (although there may be multiple cycles operating within the village landscape). So the village would be the ideal unit for applying the above formula. Unfortunately, these boundaries have not been mapped by the government. We therefore could estimate the fallow period only as an average for an entire CRD block, each of which comprised of several hundred villages. "*n*" is 2 years for most of the study area (the assumption being that the cultivation cycles of all the villages within a block are contained within the block boundary). The need for these assumptions arises because of the lack of village-level maps from most parts of Northeast India that disallow estimation of fallow periods at that level.

The literature hypothesizes that declines in fallow periods may be driven by population growth [78] or the intrusion of horticultural plantations into shifting agricultural landscapes, thereby reducing the land available for shifting cultivation and shortening fallow periods. The relevant variables would not be absolute population or plantation area, but these values normalized in some fashion. We used population values for each CRD block from the 2011 census and normalized them to population density by dividing with the geographical area of the block. For horticultural plantations, we normalized the area under horticultural plantations with the total hill area in the block, which was the total geographical area minus wet rice (valley) cultivation and water bodies, since the hill area is potentially available for plantations and for shifting cultivation. Our assumption is that population pressure acts on all land including valley cultivation, whereas horticultural plantations only compete with shifting cultivation in hilly lands.

## 5. Results

We present the results in four stages: the overall land-use map, estimates of its accuracy, the land-use statistics emerging from it, the fallow period (or land-use intensity) estimation for different parts of the district, and the patterns in and correlates of land-use intensity in shifting cultivation.

*5.1. Land-Uses in West Garo Hills District*

The land-use map of West Garo Hills for 2013–2014 as derived from this study is presented in Figure 4. Two important points stand out. Firstly, both shifting cultivation and horticultural plantations are ubiquitous and found across the district. Secondly, the map shows that the area of old-growth forest is largely located in the eastern portion, corresponding to Nokrek Biosphere Reserve in the upland region. Much of the fallows—young and old—are also found in this region, indicating longer fallow periods. Interestingly, none of the plantation crops are found in the upland reaches in the central-eastern areas of the district. Field visits confirm that the low temperatures and high humidity conditions of the region disallow growth of cashew and rubber, although areca is seen in some areas. The spread of areca is concentrated along the main roads because areca nuts can be easily collected and sent to markets. Cashew has a more even spread. Wet rice cultivation is confined to the valleys, especially the floodplains of the tributaries of the river Brahmaputra towards the western part of the landscape. The highly interspersed nature of land-uses is indicative of the predominance of smallholder agriculture.

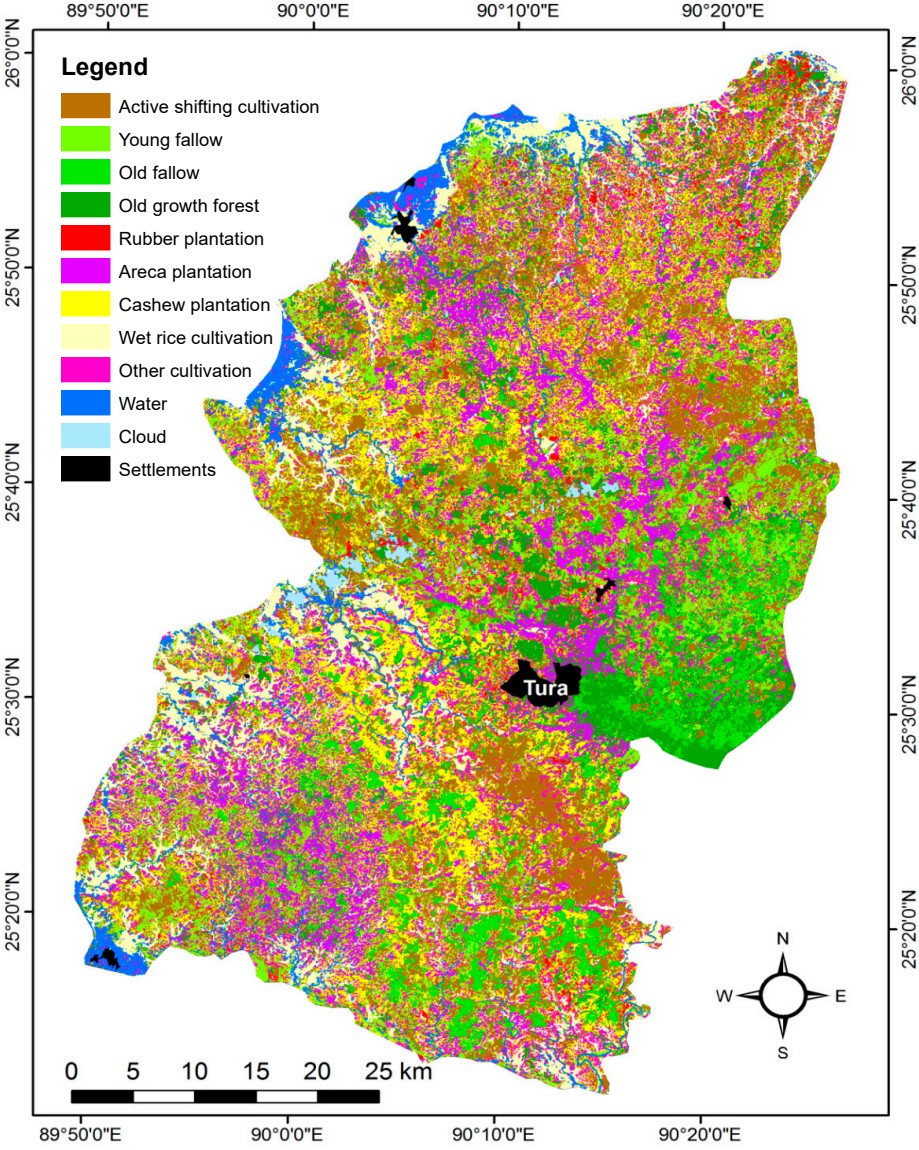

**Figure 4.** Land-use map of erstwhile West Garo Hills district in 2013–2014.

*5.2. Map Accuracy*

The 13-class map classification yielded an overall accuracy of 80% and kappa index of 0.71 (Table 1), which is quite reasonable for the mapping of landscapes of this complexity and high interspersion of classes [79]. The following features stand out:

(1) The aggregated active shifting cultivation class gets classified very well, with 89% and 91% accuracy. As the sub-matrix for the sub-classes of active cultivation shows (Table 1), there is some confusion amongst them, but aggregation improves the overall accuracy of this class.

(2) Wet rice cultivation is easily discriminated from shifting cultivation and other classes (83% and 98% accuracy). Unlike in other studies [45], no slope information had to be added to make this possible.

(3) The forest-like horticultural plantations are also identified fairly well. Amongst these, rubber and cashew are identified well (rubber: 93%; cashew: 78% and 89%), while areca is mapped with much lower user's accuracy (58% and 89%) because of the occasional confusion with old-growth forest, fallow component of 0th year active shifting cultivation classes, as well as other plantation classes. Plantation area, especially under areca and rubber, is likely to be an underestimate since many fields with young plantation saplings are sparsely vegetated and can be confused with young fallows or maybe even 2nd year active shifting cultivation areas.

(4) The fallows—young and old—are discriminated with limited accuracy (50% or lower), although the producer's accuracy for old fallow is quite high (89%). Young fallows include the 2nd year fields that were cultivated and harvested in November and then fallowed, where the signature is changing within the year and hence creating confusion with other categories. Old fallows, not surprisingly, get confused with forest, but also with young fallows—an indication of the fluidity or diversity in the fallow category. The confusion with fallows could have been avoided if the analysis was carried out in a single agricultural year. But lack of cloud-free imagery from post-clearance/post-burn (summer) period of 2013 and post-harvest period of 2014 made that impossible.

(5) The classification accuracy of old-growth forest is 58% (user's accuracy) and 66% (producer's accuracy), which is moderate. Confusion with the older fallows and young fallows is the primary reason. Shifting cultivation landscape are active production landscapes and hence old-growth forests are essentially relatively undisturbed or mature secondary regrowth forest and are occasionally used for bamboo and timber extraction for construction that creates canopy openings making them resemble fallows. Conversely, fallows contain several trees that are actively planted and that tend to make fallows resemble older forests in satellite imagery.

**Table 1.** Error matrix for the 11-class land-use map (with three active shifting cultivation land-cover sub-classes) of West Garo Hills district.

| Classified Data | Ground Data (#Pixels) | | | | | | | | | | | | | | Total Pixels | User's Accuracy(%) |
|---|---|---|---|---|---|---|---|---|---|---|---|---|---|---|---|---|
| | ASC-1Y2Y | ASC-0Y Cleared | ASC-0Y Burned | ASC | YF | OF | OGF | RP | AP | CP | WRC | OC | Wt | Cl | | |
| ASC-1Y2Y | 976 | 7 | 0 | 983 | 126 | 0 | 0 | 2 | 0 | 5 | 1 | 1 | 0 | 0 | 1118 | 87.3 |
| ASC-0Ycleared | 56 | 1096 | 90 | 1242 | 74 | 0 | 0 | 0 | 0 | 0 | 10 | 5 | 0 | 0 | 1331 | 82.3 |
| ASC-0Yburned | 0 | 114 | 572 | 686 | 51 | 1 | 0 | 0 | 0 | 0 | 0 | 4 | 0 | 0 | 742 | 77.1 |
| ASC | 1032 | 1217 | 662 | 2911 | 251 | 1 | 0 | 2 | 0 | 5 | 11 | 10 | 0 | 0 | 3191 | 91.2 |
| YF | 148 | 55 | 20 | 223 | 251 | 0 | 1 | 4 | 1 | 1 | 2 | 0 | 0 | 0 | 483 | 52.0 |
| OF | 29 | 0 | 3 | 32 | 94 | 138 | 158 | 0 | 0 | 0 | 0 | 0 | 0 | 0 | 422 | 32.7 |
| OGF | 45 | 1 | 2 | 48 | 58 | 16 | 256 | 1 | 0 | 3 | 0 | 4 | 0 | 0 | 386 | 66.3 |
| RP | 0 | 0 | 0 | 0 | 16 | 0 | 1 | 221 | 0 | 1 | 0 | 0 | 0 | 0 | 239 | 92.5 |
| AP | 0 | 13 | 10 | 23 | 7 | 0 | 13 | 0 | 67 | 4 | 0 | 2 | 0 | 0 | 116 | 57.8 |
| CP | 0 | 0 | 1 | 1 | 21 | 0 | 11 | 3 | 5 | 147 | 0 | 1 | 0 | 0 | 189 | 77.8 |
| WRC | 2 | 4 | 0 | 6 | 1 | 0 | 1 | 0 | 0 | 0 | 389 | 0 | 0 | 0 | 397 | 98.0 |
| OC | 0 | 8 | 8 | 16 | 21 | 0 | 1 | 0 | 1 | 0 | 35 | 20 | 2 | 0 | 96 | 20.8 |
| Wt | 2 | 1 | 0 | 3 | 0 | 0 | 1 | 0 | 1 | 0 | 30 | 0 | 155 | 0 | 190 | 81.6 |
| Cl | 2 | 0 | 0 | 2 | 0 | 0 | 0 | 7 | 0 | 4 | 0 | 0 | 0 | 86 | 99 | 86.9 |
| Total pixels | 1260 | 1299 | 706 | 3265 | 720 | 155 | 443 | 238 | 75 | 165 | 467 | 37 | 157 | 86 | 5808 | |
| Producer's Accuracy (%) | 77.5 | 84.4 | 81.0 | 89.2 | 34.9 | 89.0 | 57.8 | 92.9 | 89.3 | 89.1 | 83.3 | 54.1 | 98.7 | 100.0 | | |

Overall accuracy: 79.9%

Kappa index: 0.71

Active shifting cultivation (ASC) land-cover classes: ASC-1Y2Y = 1st year harvested field to 2nd year cleared field; ASC-0Ycleared = Fallow to 0th year cleared field; ASC-0Yburned = Fallow to 0th year burned field; Land-use classes: ASC = All active shifting cultivation land-covers combined; YF: = Young fallow; OF = Old fallow; OGF = Old-growth forest; RP = Rubber plantation; AP = Areca plantation; CP = Cashew plantation; WRC = Wet rice cultivation; OC = Other cultivation; Wt = Water; Cl = Cloud.

*5.3. Land–Use Extents*

Table 2 provides the land-use extents of all classes mapped. Active shifting cultivation alone covers 612 km$^2$ or approximately 18.2% of the landscape in West Garo Hills district. When combined with young and old fallows, the total area under the shifting cultivation cycle is about 39% (1306 km$^2$), making it the single largest land-use. Second, horticultural plantations as a whole are also a large proportion (30%) of the landscape, with areca and cashew being the dominant types. This reflects the early and substantial penetration of these two plantation crops in this region, a fact supported by key informant interviews and official agricultural reports. Third, old-growth forest is only 9.7% of the district (327 km$^2$) and is mostly restricted to the Nokrek Biosphere Reserve in the eastern highlands and in a few gorges and hilltops. Fourth, the area under wet rice cultivation is 8.5% or about half of the area under active shifting cultivation, making it also a significant contributor to livelihoods and diets not only in the lowlands (especially in the western parts of the district where the hilly terrain opens out as it merges with the Brahmaputra floodplains), but also up in the hills where they undertake rice cultivation in small stretches in the valleys between hills.

**Table 2.** Area under different land-uses for West Garo Hills district as per two-season classification.

| Class Name | Area (sq. km) | Area (%) |
|---|---|---|
| Active shifting cultivation | 612 | 18.2 |
| Young fallow | 483 | 14.3 |
| Old fallow | 211 | 6.3 |
| Old-growth forest | 327 | 9.7 |
| Rubber plantation | 114 | 3.4 |
| Areca palm plantation | 446 | 13.2 |
| Cashew plantation | 443 | 13.1 |
| Wet rice cultivation | 287 | 8.5 |
| Other cultivation | 248 | 7.4 |
| Water | 168 | 5.0 |
| Cloud | 32 | 0.9 |
| Totals | 3371 | 100 |

*5.4. Patterns in, and Correlates of Fallow Periods*

The ratio of the aggregate area of fallows and active shifting cultivation (ASC) for the district as a whole i.e., the district-level F:ASC ratio, is 1.2. Multiplying this by 2 (the typical period of ASC of a particular patch: see Section 4.2.4.), we get the average fallow period for the entire district as 2.4 years. This indicates a very intensive shifting cultivation system on the whole. We then similarly estimated the average fallow periods for each CRD block of the district (Table 3). The average block-level fallow period was lowest in Dadenggiri (1.4 years) and the highest in Rongram (4 years). Clearly, there is some variation within the district, but even four years is a very short fallow period.

**Table 3.** Estimation of fallow periods (in years) in different CRD blocks of West Garo Hills district.

| Community & Rural Development (CRD) Block | Fallow:Active Shifting Cultivation (F:ASC) Ratio | Fallow Period (with $n = 2$) |
|---|---|---|
| Dadenggiri | 0.7 | 1.4 |
| Selsella | 0.7 | 1.5 |
| Gambegre | 0.8 | 1.5 |
| Tikrikilla | 1.0 | 2.1 |
| Dalu | 1.4 | 2.7 |
| Betasing | 1.5 | 2.9 |
| Zikzak | 1.9 | 3.8 |
| Rongram | 2.0 | 4.0 |
| West Garo Hills district | 1.2 | 2.4 |

Tree plantation crops are very commonly planted by farmers in ASC fields to expand livelihood options via land-use conversion. So to see whether expanding plantation areas may be leading to the reduced area for shifting cultivation and thereby reducing the fallow period, we chose the two most accurately mapped plantation types, viz., cashew and rubber, and examined the relationship between the plantation-hill area ratios and the F:ASC ratios in CRD blocks. We noticed a negative correlation between these two variables (Spearman's rho = −0.27; Sig. = 0.2, 1-tailed), indicating that fallow periods shorten where the area under tree plantation crops is higher (Figure 5).

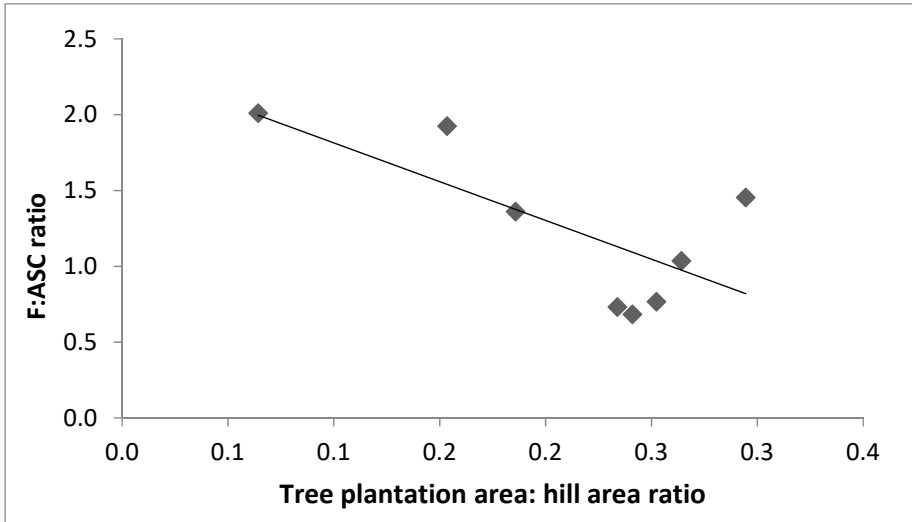

**Figure 5.** Relationship between tree plantation to hill area ratios and fallow to active shifting cultivation ratios (F:ASC) in the CRD blocks of West Garo Hills district.

Similarly, we examined the correlation between population pressure and the F:ASC ratio. Our analysis (Figure 6) found no significant relationship (Spearman's rho = −0.09; Sig. = 0.4, 1-tailed). Ideally, we should have combined both explanatory variables (population density and plantation area fraction) into a single multiple regression, but this was not possible because of the small sample sizes (*n* = 8). So our results must be treated as indicative rather than conclusive. They are, however, supported by field data gathered through discussions with villagers.

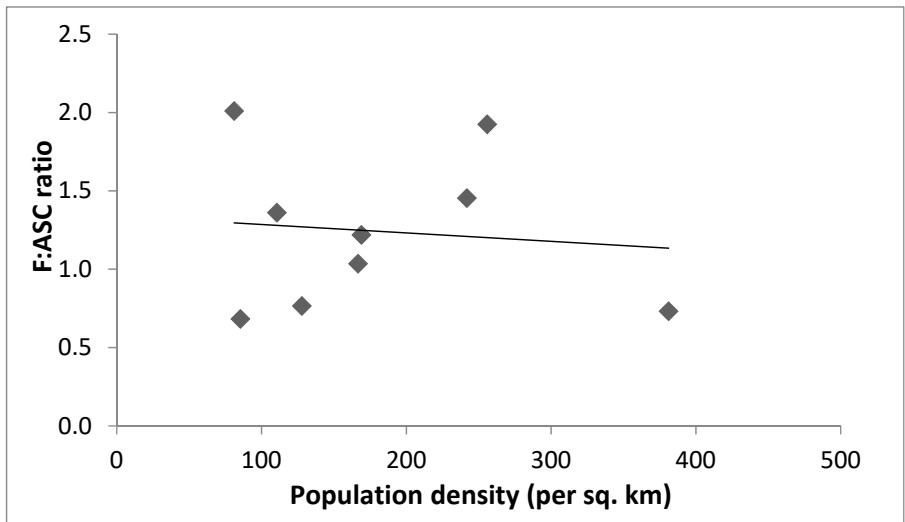

**Figure 6.** Relationship between population density and fallow to active shifting cultivation (F:ASC) ratios in the CRD blocks of West Garo Hills district.

## 6. Discussion

The above results have significance and implications for several ongoing policy debates: the extent of shifting cultivation (from the perspective of agricultural development), the extent of forest cover in Northeast India (from a conservation perspective), and the intensity of cultivation and the possible role of new crops in it. They also have wider methodological implications for the classification and estimation of shifting cultivation. We discuss each of these aspects below.

### 6.1. Occulted Farms

Our results suggest that shifting cultivation, including its active shifting cultivation and fallow phases, is the single biggest land-use in the region. To place this finding in context, we compared our estimates of the extent of shifting cultivation with estimate from several other sources (Table 4). Four sources provide data for the roughly same region (district or at least a three-district cluster) as our study area: three based on remote sensing, viz., The Wastelands Atlas of India 2011 by NRSC-MRD [67], Talukdar, Ghosh and Roy [21], Sarma et al. [69], and one from conventional bottom-up administrative reporting to the Directorate of Economics and Statistics (DES) of the Ministry of Agriculture [80]. We observe several discrepancies. Firstly, most studies do not have a well-separated class for active shifting cultivation; they refer to it using a multiplicity of terms such as barren and uncultivable land (DES) or current jhum and abandoned jhum without clarifying what land-covers they contain [21]. Otherwise, it is only found in the Wastelands Atlas as one of the categories of wasteland, pointing to an explicitly pejorative perspective about shifting cultivation. Other academic studies [22] also use such disparaging nomenclature while mapping land-cover/land-use in Northeast India. Overall, the lack of understanding of this land-use and bias about shifting cultivation are apparent.

Secondly, regardless of definitions used, all other remote-sensing based estimates are far below our estimate of active shifting cultivation (18.2%), ranging from 2–3% [67] to 6% [21]. While the estimates do not pertain to the same year, this cannot explain this major difference. Considering that NRSC-MRD [67] estimates are based on satellite image analysis, and since they have mapped both cleared and burned fields (but not 2nd year fields), their estimate of active shifting cultivation should be near half of ours (i.e., 336 km$^2$), but instead is only 115 km$^2$ casting serious doubt on their methodology and quality of interpretation. Similarly, Sarma et al.'s [69] estimates of both cultivation and fallow are simply too low, likely because of a weak methodology. Talukdar et al.'s [21] estimates are not for the same period or region as ours, but their total estimate of active shifting cultivation of 500 km$^2$ (6%) for the three Garo Hills districts is smaller than ours for a single district. Fieldwork and literature indicate that the only noticeable change in the three Garo Hills districts in the last 10–15 years has been a possible decline of shifting cultivation area because of the rise in plantations. Thus, studies conducted earlier than ours should, if at all, report a larger extent under active shifting cultivation. The underestimation reinforces policy blindness to and the bias against shifting cultivation.

Thirdly, DES' bottom-up estimates are also completely wrong, but for a different reason. While administratively assembled land-use data are known to be somewhat inaccurate globally, the bigger factor here seems to be that DES imposes a uniform classification across the country, which fits regions with settled agriculture (where categories such as Current fallows, Fallow lands other than current fallows and Cultivable wasteland are relevant). This classification is, however, not at all suitable for shifting cultivation landscapes. When combined with the bias visible in its categorization as a type of wasteland, it appears that the fundamental problem is not estimation or mapping methods, but a refusal to acknowledge it as a distinct and legitimate agricultural land-use relevant for people's livelihoods.

Finally, neither government reports nor most remote-sensing studies (except Roy et al. [66]) identify or systematically estimate the area under plantations, in spite of their significant share of the landscape (~30%). In the remote-sensing studies, plantations have, in all likelihood, been merged with forest. But in DES data, which are supposedly collected bottom-up, only areca is reported, while cashew (which is equally pervasive) and new entrants like rubber are omitted. The area under areca plantations is itself grossly underestimated. This absence of the largely monoculture horticultural

plantations in the mapping exercises and its underestimation in government statistics is a serious flaw, considering how significant tree plantations like cashew and areca are from agricultural/rural livelihood [1,81] and environmental perspectives [4,82]. It reinforces the impression of a region with static agricultural practices, when in fact the region is undergoing multi-faceted changes.

**Table 4.** Comparative extent of shifting cultivation and all agriculture land uses (in km$^2$) as recorded by our study and other studies.

| Source | | This Study | DES, Dept. Agriculture, Govt. India 2013–2014 | NRSC-MRD 2011 (The Wastelands Atlas of India) | Talukdar et al. 2004 | Sarma et al. 2015 |
|---|---|---|---|---|---|---|
| Year of data collection | | 2013–2014 | 2013–2014 | 2008–2009 | 2000 | 2013 |
| Spatial scale | | One district (West Garo hills) | | | Three districts (Garo hills region) | |
| Active Cultivation (without tree canopy cover) | Wet rice cultivation | 287 (8.5%) | | | 745 (9%) | |
| | Active shifting cultivation | 612 (18.2%) | 72 (2%) [1] | 115 (3%) | 500 (6%) [4] | 159 (2%) |
| Fallow agricultural land | Fallow (young and old) | 694 (20.6%) | 606 (16%) [2] | 463 (12.5%) [3] | 4112 (50%) | 43 (0.5%) |
| Tree-like agriculture | Plantations | 1003 (29.8%) | Areca = 167 (5%) | | | |
| | Other cultivation | 248 (7.4%) | | | | |
| Total district area (km$^2$) | | 3371 | 3677 | 3714 | 8167 | 8167 |

Note: Area figures are rounded off to the closest whole number. The district area estimated in our study is smaller than that quoted in The Wastelands Atlas of India 2011 and Directorate of Economics and Statistics (DES) 2012–2013 due to variations in boundaries that could not be sorted out. The difference in total area, however, does not explain the variation in individual land-use estimates. [1] DES class 'barren and uncultivable land' comes nearest to active shifting cultivation. [2] Includes DES categories 'current fallows', 'fallow lands other than current fallows', 'cultivable wasteland', and 'land with open scrub'. Additional data gathered from DES office in Meghalaya. [3] The standard terminology for Fallow category is 'abandoned jhum' in the Atlas. [4] Classified as 'current jhum' but unclear which land-covers this class contains.

*6.2. Imagined Forests and Spurious Deforestation*

While shifting cultivation and even horticulture are rendered invisible, forest cover seems to be overestimated in most studies and reports. A comparison of our estimates for natural or old-growth forest with other estimates is given in Table 5. We have included only those sources that provide, or where we could generate by clipping their maps, district-level data. Compared to our estimate of about 10% old-growth forest, other estimates range from 45% (DES) and 58% by Roy et al. [66] to 79% by FSI [62].

The reasons for this over-estimation are two-fold. First, there is the recurrent problem of definition. Forest Survey of India (FSI) defines forest as any area with a tree canopy density >10%. Its estimate of forest cover therefore includes all forest-like classes such as horticultural plantations and older shifting cultivation fallows. Wadsworth and Lebbie [83] highlight similar problems in the context of forest inventorying in Sierra Leone. DES' definition is even more problematic: "any legal enactment dealing with forest or administered as forest whether state owned or private, and whether wooden or maintained as potential forest land. The area of crops raised in the forest and grazing lands or areas open for grazing within forests should remain included under forest area." This means DES does not report actual land-use or land-cover, only its legal status. Both approaches provide a misleading picture of the status of natural or uncultivated vegetation in the region. Roy et al. [66] has better definitions than other studies but their land-use map data we obtained from the ORNL is too coarse-grained for a highly fragmented smallholder farm landscape with small-sized fields, casting doubt on the quality of data collection.

**Table 5.** Comparative extent of all forest-like land uses (in km$^2$), as recorded by our study and other studies.

| Source | | This Study | FSI 2015 [62] | DES 2013–2014 [80] | Roy et al. 2015 [66] [1] |
|---|---|---|---|---|---|
| Year of data collection | | 2013–2014 | 2013–2014 | 2013–2014 | 2005 |
| Scale | | West Garo hills district | | | |
| Tree cover/'Forest-like' land-use | Old-growth forest | 327 (9.7%) | 1541 (46%) | 1647 (45%) | 2202 (58%) |
| | Mono-species tree plantations | 1003 (29.8%) | | Areca nut = 167 (5%) | 377 (9%) |
| | Old Fallow | 211 (6.3%) | | 606 (16%) | |
| Non-forest | Other cultivation | 248 (7.4%) | 727 (20%) | | 137 (4%) |
| | Water, cloud | 200 (5.9%) | | | |
| Total district area (km$^2$) | | 3371 | 3715 | 3677 | 3820 |

Note: the district area estimated in our study is smaller than that quoted in FSI studies due to variations in administrative boundary data layers that could not be corrected. The extent of variation in total area, however, does not explain the variation in individual land-use estimates. All figures are rounded off to make for easy comparison.
[1] The deposited classified land-use map for 2005 was obtained from Oak Ridge National Laboratory (ORNL DAAC): https://daac.ornl.gov/VEGETATION/guides/Decadal_LULC_India.html.

The second issue is the quality of interpretation. Even if we sum up the area under all tree-like classes (old-growth, horticultural plantations, and old fallows), we get only 46% as against FSI's 79%. A closer examination of FSI's maps suggests that horticultural plantations are included as forests because of their tree canopy cover, but that they may have confused ASC and young fallows with FSI's scrub class, and also incorrectly identified many classes as one or the other forest class. Figure 7 shows a close-up of an area of our land-use map and the corresponding region mapped by the FSI for the same period displaying the gross error in interpretation of plantations as forest. Since FSI does not provide details of their ground data collection and validation strategy, it is not clear whether this could be due to inadequate ground truth or uncertainties in particular classes. In the case of Roy, et al. [66], although their classification is for the year 2005, our discussions in the field suggest that the forest landscape has not changed dramatically in the 2005–2015 period. While the presence of a separate plantation class means that there is no definitional issue, they nevertheless overestimate the area under natural forest: 58% as against our 16% (including old fallows). The possible explanation of the overestimation in both cases is that both being all-India mapping exercises, they are unable to devote adequate effort to individual regions with distinct geographies and land-use systems to map them accurately. This calls for the need to re-look at such large mapping exercises, both in their conceptualization of categories and their execution of ground data collection.

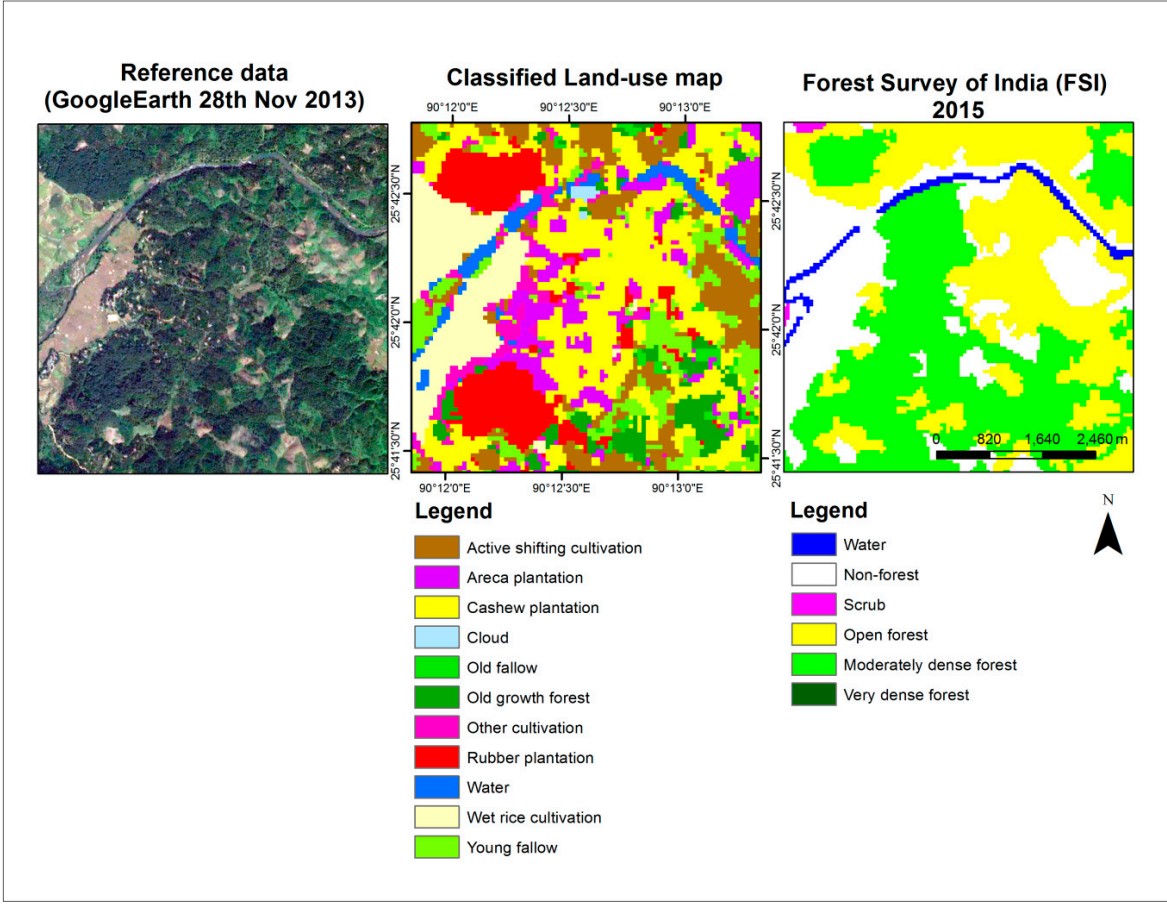

**Figure 7.** Map panel of our classified land-use map (2013–2014) and Forest Survey of India (FSI) map showing misidentification of plantation classes as forest classes by FSI in their The State of Forest Report 2015. The FSI data collection period matches with ours.

Our estimates, though for only one district, call into serious question claims about the Northeast Indian region housing a quarter of India's forest cover [62]. Our mapping not only demonstrates how widespread shifting cultivation is, but also uncovers major land-uses, viz. areca, cashew and rubber plantations that now occupies almost a third of the landscape. By conflating horticultural plantations with forests (FSI) or largely overlooking them (DES), state agencies are undoubtedly doing a dis-service to policy-makers and anyone who seeks to understand the ecology or livelihoods of the region. Accurately defining and delineating the different plantation crops is essential for land-use mapping not only to avoid conflation of other annual/perennial crop area or old-growth forest with tree plantations, but also because the type of tree plantation could also have different repercussions for livelihoods, biodiversity, carbon sequestration, hydrological services, and their trade-offs. For example, pine and teak plantations in some areas are seen as beneficial [84], while industrial scale palm oil and rubber expansion is seen as contributing negatively to both biodiversity as well as livelihoods [44,85]. Explicating this multi-dimensionality is what provides land-use mapping with its public value. Similarly, overlooking the fact that secondary forest is integral to the shifting cultivation cycle and not separately defining and mapping shifting cultivation fallows prevents a nuanced understanding of the complex mosaic of vegetation that exists on this landscape and its implications for biodiversity, carbon and livelihoods.

Several studies also infer that deforestation is taking place and attribute it to shifting cultivation in the Garo Hills. But those that have used time-series data on forest cover [22,63,66] do not have shifting cultivation as a class, so it is not clear how the causality is inferred. The nature of shifting cultivation is such that, over roughly 10-years, some areas would be deforested (when converted to

active shifting cultivation), while other areas would transit into secondary forest (when fallowed). As a result, concluding that shifting cultivation causes deforestation requires a) going beyond the individual plot or pixel and looking at the whole landscape to see if the area under the shifting cultivation cycle has indeed expanded into the hitherto undisturbed forest, and b) being rigorous in identifying this hitherto undisturbed forest. None of the studies that blame shifting cultivation for deforestation have done this.

### 6.3. Fallow Periods and Correlates of Fallow Land-Use Intensity

Our separation of fallows from active shifting cultivation and old-growth forest enables us to estimate the average duration of the fallow period. Although the method is approximate and there is variation within the district, the fallow period is clearly rather short (1–4 years), which matches the central tendency of intensification noticed in studies globally [2]. Evidently, this shifting cultivation landscape is undergoing major transformations with the adoption of horticultural tree plantations. Our exploration of the possible correlates of changes in fallow duration suggests that the expansion of area under horticultural plantations may be a significant driver of this transformation. This is supported by field data. Field observations indicated that saplings of cashew, areca and rubber are directly planted into the ash-filled 0th year burned cultivation field along with the annual crops usually found in the shifting cultivation fields. After the 2nd year harvest, the field undergoes land-use conversion to become a plantation field. This effectively reduces the area under fallow and the area that would be available for the next round of active shifting cultivation. Field visits also confirmed that plantation extent was lesser in the higher elevation eastern CRD block of Rongram where none of three crops grew well, and consequently, fallow periods were longer there.

At the same time, the attachment to shifting cultivation continues for a variety of poorly understood reasons. Globally too, while there is general agreement that long fallow systems are more productive than short fallow systems [74], communities continue to practice shifting cultivation in the shrunken area [1,86]. The governments in Northeast India (and elsewhere), having assumed that shifting cultivation is harmful, have initiated a variety of jhum control schemes, prominent among which is the introduction of horticultural plantations to replace shifting cultivation [18]. However, it is not clear how horticulture will meet food and nutritional needs, and whether dependence upon markets or public distribution systems for these needs (and concomitantly reduced self-sufficiency) is desirable. Investing in a more nuanced understanding of the reasons for both the adoption of horticultural crops and the continued attachment to shifting cultivation, as well the possible consequences of reduced fallow periods in particular contexts, would be preferable to an exclusive policy orientation on removing shifting cultivation and expanding plantations as has thus far been attempted in Northeast India.

### 6.4. Methodological Issues in Mapping Complex Shifting Cultivation Landscapes

Our results also have wider implications for mapping and how the mapping of dynamic shifting cultivation landscapes in tropical regions needs to be carried out. First, the results highlight the need for care in defining land-use categories. For those engaging in land-use mapping in general, our results point to need to recognize shifting cultivation as a distinct land-use class, failing which active shifting cultivation gets mis-classified as scrub, grassland, or barren, and the fallows get lost in 'degraded forest' or 'secondary forest'.

Second, even for those engaged explicitly in mapping shifting cultivation, clarity on whether one is mapping only active shifting cultivation or the whole cycle, and the nature of land-covers in different seasons and different phases of active shifting cultivation and fallow, as outlined in Figure 1 for our region, is essential. As mentioned in Section 2.2, mapping based on either the cleared fields from the post-clearance or post-harvest season [45,46] or only burned fields [50,51] risks providing an underestimate of the extent of active shifting cultivation, as they may miss out plots in their second or subsequent years of active shifting cultivation. Depending on the region, shifting cultivation goes through 1–5 years of cultivation on the same field before the field is left fallow [13,74]. Cultivation fields

beyond the first year often have some perennial crops and are filled with grass and weeds that influence spectral signatures of these fields. Most remote sensing studies avoid these land-cover distinctions that differentially determine signatures and influence shifting cultivation estimates. Excessive reliance on already mapped products (with embedded class definition issues), visual interpretation of land-uses, and use of sub-products of remotely sensed data, i.e., vegetation indices, burned area ratios, etc. for data analysis at the cost of sound field knowledge and data collection, could be another reason for this.

Thirdly, estimating the period is important for understanding the sustainability of production, the processes of intensification and the implications for conservation. While using multi-year data for estimating how long a patch remains fallow may be best, such an approach may be quite laborious. Our approach of estimating the fallow period from one-time data using the F:ASC ratio and the duration of active shifting cultivation of each plot provides a quick preliminary estimate. If it can be coupled with information on the micro-level (say village-level) boundaries within which the cultivation is cycling, it can provide a spatially disaggregated set of estimates, with which the association of other spatial and non-spatial variables can analyzed.

## 7. Conclusions

We adopted an approach involving carefully delineating the shifting cultivation cycle, its land-cover forms and identifying other policy and socially relevant land-use categories. We then devised image selection and interpretation strategies suitable for discriminating these land-use classes and supported them with extensive ground data for training and validation. This enabled us to generate a reasonably accurate land-use map that throws important light on the extent, intensity and distribution of shifting cultivation (active shifting cultivation and fallow phases) and other land-uses in the study region in Northeast India. We also devised a strategy to estimate the fallow period using the fallow area to active shifting cultivation area ratios and used the estimates for different sub-regions to explored possible association with two likely drivers of the reduction in fallow periods (expansion in horticultural plantations and population growth). We believe this is one of the first efforts to comprehensively map and understand the complex set of land-uses prevailing in Northeast India at this scale, detail and accuracy.

Empirically, our findings regarding the large extent of shifting cultivation and equally significant extent of horticultural plantations are in sharp contrast to existing government statistics and reports that portray this region as heavily forested, with shifting cultivation variously seen as being insignificant, simply a wasteland, and being a major cause of deforestation. Our findings assume a special significance in the light of a recent report by the highest planning body in India (the NITI Aayog) that criticizes the inconsistent estimates of the extent of shifting cultivation in Northeast India and calls for an urgent need for authentic estimates of shifting cultivation [87]. Our study contains the essential framework required for creating such a database and it is hoped that studies such as this one will pave the way for accurate information on shifting cultivation that the NITI Aayog demands. In the absence of such information, governments will push for potentially harmful policies such as promoting even more monoculture horticultural plantations in an already changing landscape or focusing on top-down forest conservation rather than bottom-up agro-diversity and biodiversity conservation. Further, although our study is limited to one district, field observations, discussions with experts and studies such as Roy and Joshi [71] indicate that shifting cultivation is a major land-use across Northeast India, and plantations of cashew, areca and others like rubber and oil palm have also expanded enormously across the region. Policy makers and mapping agencies in India need to recognize this reality that is transforming not only landscapes but societies as well [88,89] and as a result modify official reporting protocols to accurately capture these land-use forms.

Conceptually, our work contributes to the promising new discourse on the normative aspect of mapping and land system science [90]. We emphasize the need to explicitly choose socially relevant land-use classes as the focus of mapping, rather than classes for which one may have some implicit preference, or (land-cover) classes that are easily separable in satellite images. While this is especially

important from the point of view of contributions of mapping for policy and societal transformation, it is of course easier said than done. The challenge of mapping shifting cultivation epitomizes the challenge of distinguishing land-use from land-cover that confronts all satellite imagery-based mapping efforts. The complex set of land-covers that make up different phases of shifting cultivation demand a contextual understanding of the practice and the vegetation forms it creates, which can only come from extensive field work. They also require a clearer definition of what one is trying to map: newly burned fields, or active shifting cultivation, or total cultivation, and so on. Clearer definition and discrimination will also enable getting at the important question of shifting cultivation intensity or fallow period and its dynamics in space and time, and contribute to our understanding of this complex and dynamic land-use form.

**Supplementary Materials:** The following are available online at http://www.mdpi.com/2073-445X/8/9/133/s1, Figure S1: Differences in spectral signature of the two year-long different land-use classes (showing standard error around the mean of the DN value for ground data pixels) in the Landsat 8 OLI sensor in the two-season stacked image corresponding to the post-harvest and post-clearance/burn data collection periods (ASC=active shifting cultivation); Table S1: Land-cover classes and their description, local names used in Garo, and the corresponding land-use classes used in the classification; Table S2: Number of ground data polygons and corresponding pixels used for training and accuracy assessment.

**Author Contributions:** A.J.K. and S.L. conceptualized the research and methodology. A.J.K., S.L. and H.N. conceived and designed the formal analysis. A.J.K. aided by S.L. acquired grant for fieldwork; A.J.K. conducted fieldwork and classification of remote sensed data. A.J.K., S.L. & H.N. co-wrote the paper.

**Funding:** The first author received Ph.D. research support from the Jamshedji Tata Trust, Tata Social Welfare Trust, and a Fellowship grant from Duleep Matthai Nature Conservation Trust (August 2013–2015) that enabled carrying out fieldwork.

**Acknowledgments:** We acknowledge with gratitude the support of the following institutions and individuals: The authors are grateful for the institutional support from the Academy for Conservation Science and Sustainability Studies at the Ashoka Trust for Research in Ecology and the Environment (ATREE) during the course of research; to the Ecoinformatics lab associates at ATREE for technical support. We thank the villagers in West Garo Hills who helped the first author understand the agricultural and related land-use systems in the many villages visited; B. K. Tiwari, Environment Studies department, North-Eastern Hill University (NEHU), and Daniel Ingty at the Meghalaya Basin Development Authority (MBDA) for assistance and facilitation during the first author's fieldwork; Mahnseng Momin of the A·chik Evangelical Association for providing office space and internet facility to use GoogleEarth during fieldwork, and Niksamseng Marak and Anaseng Momin for language assistance during fieldwork. We would also like to thank the three reviewers of our paper for their detailed comments that greatly improved the manuscript.

**Conflicts of Interest:** The authors declare no conflict of interest. The funders had no role in the design of the study; in the collection, analyses, or interpretation of data; in the writing of the manuscript, or in the decision to publish the results.

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
