# Peer review of "Farms or Forests? Understanding and Mapping Shifting Cultivation Using the Case Study of West Garo Hills, India"

_land, doi:10.3390/land8090133_

Round 1
Reviewer 1 Report
some very general comments to consider:
59: besides duration of fallow the practice is importance – i.e. with or without burning, slash and burn or slash and mulch
730: you might wish to refer to another alternative, which is Conservation Agriculture, and which is defined as “sustainable intensification”, in which it halts erosion and soil degradation by no-till and no-burning, it protects soil by soil cover, and it enhances soil health by diversified cropping systems including cover crops, as such intensifying the “fallow” periods of shifting cultivation and enabling a sustainable production system without need of clearing new land.
Author Response
We thank the reviewer for the comments provided. Since overall structure and rigor of the paper is indicated as good, we proceed with our responses to more detailed comments provided by them. The comments are provided.
Please see the attachment.

Reviewer 2 Report
Line 16 - As I understand from this phrase, this paper is a methodological one, and you focus on a specific methodology to assess and understand the shift between farms and forests. In the paper, you mentioned, obviously, about the methods section but also you have results based on the methods and analysis.
Please, clarify this aspect and change this phrase.
Line 20 - You can mention here the objective of the paper.
Lines 24-26 - It is good that you mentioned here the results that you got the results you
achieved.
Lines 70- You mentioned the defining and analyzing land-use/land-cover based on the review analysis. This sub-section is quite large, so, you can simplify all the literature review and to include in the introduction section.
Lines 286-294 - You can include this main land-uses in a table.
Figure 3 - It is more suitable to include this foto-panel in the sub-section "4.1.2. Ground data collection".
Figure 4 - You can include additional information on the map as the name of important cities or rivers.
Table 4 - It is quite complicated to follow the table. Please, modify and find a suitable manner
to present the shifting cultivation and all agriculture land uses.
Lines 584 - "Methodological contribution" is an important part of your paper so, you must
include this into the "Results section".
Reviewer 3 Report
This paper provides an explanation of shifting cultivation, and how to improve mapping accuracies of this land use type. The authors provide a detailed background as to what shifting cultivation is. They explain their methods and how their methods are an improvement necessary to map shifting cultivation and classify it apart from old-growth forest. They then compare their results to other existing classifications, and raise some concerns about the existing land cover/land use maps policy is based on. I think this paper provides a meaningful and useful contribution to the field of land cover mapping, as well as the importance of context in providing accurate maps.
General
This paper is well written, and is a valuable contribution to the field. However, it is quite long in places. I suggest you try cut down some of the text in the introduction and discussion. Specifically, you use many short paragraphs, some of which do not flow from one to the next, and the links between them are weak (and sub-headings add to this, allowing you to not have to create that flow). Please combine paragraphs when you reduce text, and focus on creating a flow in the narrative. Another big gap I found was the distinction between old and young fallow. How exactly did you differentiate between the two in the classifications?
